# Engineering Multicolor Radiative Centers in hBN Flakes by Varying the Electron Beam Irradiation Parameters

**DOI:** 10.3390/nano13040739

**Published:** 2023-02-15

**Authors:** Federica Bianco, Emilio Corte, Sviatoslav Ditalia Tchernij, Jacopo Forneris, Filippo Fabbri

**Affiliations:** 1NEST Laboratory, Istituto Nanoscienze-CNR and Scuola Normale Superiore, Piazza San Silvestro 12, 56127 Pisa, Italy; 2Physics Department, University of Torino and Istituto Nazionale di Fisica Nucleare Sez. Torino, Via P. Giuria 11, 10125 Torino, Italy

**Keywords:** hexagonal boron nitride, electron irradiation, photoluminescence, defect-related light emission

## Abstract

Recently, hBN has become an interesting platform for quantum optics due to the peculiar defect-related luminescence properties. In this work, multicolor radiative emissions are engineered and tailored by position-controlled low-energy electron irradiation. Varying the irradiation parameters, such as the electron beam energy and/or area dose, we are able to induce light emissions at different wavelengths in the green–red range. In particular, the 10 keV and 20 keV irradiation levels induce the appearance of broad emission in the orange–red range (600–660 nm), while 15 keV gives rise to a sharp emission in the green range (535 nm). The cumulative dose density increase demonstrates the presence of a threshold value. The overcoming of the threshold, which is different for each electron beam energy level, causes the generation of non-radiative recombination pathways.

## 1. Introduction

Electron irradiation has been an interesting tool for engineering layered materials and tailoring their properties [1,2,3]. It is possible to consider two main classes of electron irradiation process: low-energy irradiation, where the electron beam energy is compatible with a scanning electron microscope (SEM) or a transmission electron microscope (TEM), i.e., 5–300 keV, and high-energy bombardment that is a process involving electrons with energy beyond 1 MeV.

Among two-dimensional (2D) materials, graphene was the first one in which electron-irradiation-induced defects were studied both experimentally and theoretically [4,5,6,7,8,9]. The interaction of the electrons with the graphene lattice can induce profound structural modifications. The graphene can suffer a decrease of its crystallinity in favor of a nanocrystalline or amorphous nature, depending on the irradiation doses. This can be obtained even at beam energy in the low-energy range [6,7,10,11,12,13]. Importantly, electron-induced defects in graphene have already demonstrated an enhancing of its chemical reactivity, leading to graphene chemical functionalization [14,15,16]. In addition, electron-irradiation-induced defects have been employed in the engineering of graphene electronic properties [17,18,19,20,21] and, more generally, in sensing applications [22,23,24,25,26,27,28,29]. In the case of high-energy electron irradiation, graphene has been mainly employed in composites for radiation shielding applications [30], therefore structural defects induced by MeV electrons in graphene sheets have been very poorly studied [31,32].

Transition metal dichalcogenides (TMDs) are another class of layered materials, where the effect of the interaction with electron beams has been widely explored for tuning their structural and optical properties [2]. In terms of structural properties, molybdenum-based TMDs, namely MoS_2_ and MoTe_2_, undergo a transition from the semiconducting phase (2H for MoS_2_ and 1H for MoTe_2_) to the metallic phase (1T for MoS_2_ and 1T′ for MoTe_2_), under electron bombardment inside a TEM [33,34,35,36]. In addition, the formation of Mo nanoparticles has been demonstrated by massive desulfurization due to high-dose electron showering of multilayer MoS_2_ flakes [37]. Meanwhile, in terms of optical properties, electron irradiation of TMDs has been used to induce below-bandgap radiative transitions, attributed to the generation of chalcogen vacancy [38,39,40]. Recently, 100 keV electron irradiation has been demonstrated to increase emission yield of quantum emitters in strained WSe_2_ sandwiched in hBN flakes, by increasing the concentration of defects [41]. In the case of high-dose [37,42] or high-energy [43] electron bombardment, the formation of non-radiative recombination pathways causes the quenching of the excitonic light emission of TMDs. 

In the case of hexagonal boron nitride (hBN), electron irradiation has been mainly used to tailor the optical properties by the controlled generation of radiative color centers. Recently, the color center generation in hBN has focused on the isolation of point defects with single photon emitter (SPE) characteristics. In particular, two recent works have demonstrated the position-controlled generation of blue-emitting (435 nm) SPE centers by low-energy electron impinging [44,45]. Meanwhile, employing low-energy electron beam irradiation in the presence of water vapor has shown the appearance of multicolor SPE centers in the 565–775 nm range [46,47]. These experiments have been carried out with electron beam energy compatible with scanning electron microscopes, i.e., 5–15 keV. 

Additionally, high-energy electron bombardment (2 MeV) has been used to demonstrate the controlled generation of color centers with SPE characteristics in multilayer hBN flakes; in this particular case, the color centers present an intense sharp emission at 590 nm [48] or a broad emission at 810 nm [49] as consequence of the formation of single boron vacancy. 

In this work, we investigate the engineering and tailoring of radiative recombination in hBN flakes by a patterned electron irradiation process. Firstly, we evaluate light emission energy and intensity of the generated color center as a function of the electron beam energy, by sub-bandgap excitation photoluminescence (PL) spectroscopy. We demonstrate that 10 keV and 20 keV beam energies cause a broad complex emission around 660 nm, while 15 keV irradiation induces the appearance of a sharp band at 535 nm. Meanwhile, the hBN crystalline disorder induced by the electron beam irradiation is evaluated by Raman spectroscopy. The eventual trade-off between the electron-energy-related penetration depth and the thickness of the hBN flakes is evaluated by means of Monte Carlo simulations of the scattering events caused by the impinging of the primary electron beam. Then, we report the effect of the interaction of a single irradiation spot and area dose using irradiation patterns, varying the distance among the different impinging sites. In this case, the monochromatic photoluminescence maps allow us to clarify the interaction of the different irradiation spots in the pattern and the threshold evaluation of the cumulative dose density, leading to possible non-radiative pathways in the highly damaged areas. 

## 2. Materials and Methods

We obtained the hBN flakes via standard micromechanical exfoliation from bulk single crystals provided by HQ Graphene. The 300 nm SiO_2_/Si exfoliation substrate was cleaned by oxygen plasma prior to deposition of the tape, while the samples were cleaned overnight in organic solvents to remove possible glue residuals.

Defects were induced in part of the hBN flakes by irradiating with electrons accelerated at different electron beam energies, varying from 10 keV up to 20 keV. The areas to be damaged were precisely targeted by a Zeiss SEM driven by a Raith pattern generator. The patterns consisted of square arrays of spots with increasing spacing. The e-beam was set to output a current of about 0.6 nA. The e-beam delivered a spot dose of ∼3.5 nC, resulting in a dwell time of 6 s. The electron beam spot size was 10 nm. Three different spot spacings were employed, namely 200 nm, 500 nm and 1000 nm, resulting in a matrix of 10 × 10, 4 × 4 and 3 × 3 irradiation sites, respectively. It is worth noting that when increasing the number of irradiation spots in an area with the same lateral extension, the dose per area unit increases drastically. The area dose, in fact, is evaluated as the single spot dose multiplied by the number of irradiation sites divided by the area of the irradiation. Therefore, the 2 μm × 2 μm irradiated areas with 200 nm, 500 nm and 1000 nm spacing had an area dose equal to 87.5 nC/μm^2^, 14 nC/μm^2^ and 7.9 nC/μm^2^, respectively. It is worth noting that prior to the irradiation of the area of interest, we carried out the electron irradiation of a sacrificial area in order to get rid of the carbon contaminant in the SEM chamber. The process was 40 min long using the following beam parameters for all used kinetic energies: beam current of 0.6 nA, dwell time of 6 s and step size of 100 nm. Raman and photoluminescence (PL) spectroscopies were carried out with a Renishaw InVia system, equipped with confocal microscope. In the case of PL measurements, a 473 nm excitation laser and a 2400 line/mm grating (spectral resolution 0.5 nm) were employed, while for Raman experiments a 532 nm excitation laser and an 1800 line/mm grating were used. The PL spectra were performed with the following parameters: excitation laser power 500 μW, acquisition time 10 s, spot size 800 nm using a 100× objective (NA = 0.85). The PL hyperspectral maps were acquired with an acquisition time of 1 s for each spot with a spot size of 500 nm × 500 nm pixel size. The Raman spectroscopic experiments were carried out increasing the excitation power to 1 mW and with an acquisition time of 1 s. The reported Raman spectra were averaged over 5 single acquisitions. The full width half maximum (FWHM) of the Raman spectrum was obtained by a Lorentzian fitting of the hBN peak. The Raman system resolution was 1 cm^−1^. 

## 3. Results

### 3.1. Electron Beam Parameters

Figure 1 clarifies the scheme of the electron irradiation experiments presented in this work. Flakes, with thickness of about 50 nm, were employed upon identification by optical contrast with respect to the SiO_2_/Si substrate [50,51] (additional details are reported in the Appendix A). 

Firstly, we evaluated the effect of electron beam energy on the engineering of radiative recombination in hBN flakes. Three electron energy values were selected for the irradiation experiments: 10 keV, 15 keV and 20 keV. The variation of the beam energy modified the depth of the generation/recombination volume. Then, we evaluated the effect of patterned irradiation, by varying the spacing among the single irradiation spots. In particular, the irradiation pattern was performed on square areas with a lateral size of 2 μm × 2 μm, with a different spacing between single irradiation spots. The optical images of the hBN flakes, superimposed on the irradiation schemes with different spacing, are shown in Appendix A. In this perspective, the interaction of the different irradiation spots was evaluated. 

Monte Carlo simulation was carried out in order to evaluate the interaction of the primary electron beam with the hBN flakes, with an average thickness of 50 nm, varying the beam energy. Figure 1b provides the depth profiles of the scattering events generating secondary electrons (SEs). As expected when increasing the beam energy, the peak of the depth distribution moved deeper into the sample. In fact, the depth profile peak was 650 nm, 1300 nm and 2100 nm for 10 keV, 15 keV and 20 keV, respectively. These values were obtained from the energy release maps reported in Appendix A and indicate that the majority of the scattering events occurred in the silicon substrate. The increase in the beam energy also led to the enlargement of the generation/recombination volume. In particular, the radius of the generation/recombination volume was 400 nm, 1 μm and 1.7 μm, for 10 keV, 15 keV and 20 keV, respectively. In addition, we also considered the scattering events giving rise to backscattered electrons (Figure 1c) because they can participate in the defect formation dynamics in 2D materials [6,7]. Similarly to what occurs to the SE scattering events, the backscattered electron (BSE) depth distribution peaked deeper in the sample, increasing the electron beam energy. The BSE depth distribution peak occurs at 250 nm, 500 nm and 850 nm for the 10 keV, 15 keV and 20 keV irradiation, respectively. It is worth noting that, although the majority of the BSE-related scattering events occur in the substrate, a significant percentage of electrons were backscattered in the first 50 nm, i.e., in the hBN flakes. The electron trajectory maps for the different electron beam energies are presented in Appendix A. 

### 3.2. Effect of the Different Electron Beam Energies

The PL spectra obtained in areas irradiated with different beam energies are shown in Figure 2a in comparison with the reference spectrum of pristine hBN (black line). The PL spectrum of the pristine hBN presents no peculiar features in the whole analyzed range. The PL spectra of the electron-irradiated SiO_2_/Si substrate are reported in Appendix A, in order to rule out any substrate-related light emission. The areas irradiated with 10 keV (red line) and 20 keV (blue line) electron beams present a broad emission in the orange–red range. In particular, the emission is composed of two main components at 610 nm and 660 nm, with a pronounced tail at the shorter wavelength side. The main difference between these two spectra is the intensity of the broad band. In fact, the 10 keV irradiation causes a PL intensity that is double the one obtained when irradiating hBN at 20 keV. The lower efficiency in emission engineering of the 20 keV irradiation process is mainly due to the deeper penetration depth of the 20 keV electron beam with respect to the 10 keV one, as demonstrated in Figure 1b,c. In fact, the energy release of the primary electron beam occurs mainly at the SiO_2_/Si interface in the case of the 20 keV beam while, in the 10 keV process, the electron energy is released, in larger part, at the hBN/SiO_2_ interface (Appendix A). 

Similarly to 10 keV and 20 keV, the 15 keV irradiated area (green line) shows a red–orange broad emission. In addition, the PL spectrum presents a sharp PL peak in the green range, namely at 535 nm, and two additional peaks at 550 nm and 575 nm. The latter peaks are attributed to photon replicas of the 535 nm zero phonon line (ZPL). Interestingly, the 15 keV irradiation is the only process to induce the appearance of the sharp 535 nm emission. The possible cause of this effect is the trade-off between the kinetic energy of electrons and their interaction with the substrate, i.e., energy release, scattering events and their depth distribution, in which the interface interaction may play a crucial role. However, it is necessary to consider the crystallographic attribution of this emission to gain a deeper insight into the effect of the 15 keV irradiation process. In order to quantify the crystalline disorder caused by the electron irradiation, Raman spectroscopy was carried out in the same areas where the PL spectra were acquired. The Raman peak that is conventionally investigated in hBN is at 1366 cm^−1^. This Raman mode is attributed to the in-plane atom vibrations (E_2g_ mode) [52,53]. The benchmark for the crystalline quality of hBN is the FWHM of this peak. The Raman spectrum (black line), reported in Figure 2b, of the pristine hBN reports an FWHM of 8.5 cm^−1^. The standard FWHM of the hBN, provided by the supplier, is ≈8.2 cm^−1^ as recently reported [54]. This agreement indicates that the exfoliation protocol, employed for the preparation of the flake, does not cause any worsening of the material crystallinity [55]. As expected, the electron irradiation dramatically affects the local crystallinity of hBN. In the case of the 10 keV process, the hBN Raman mode is completely quenched, suggesting a local amorphization of the lattice. In the case of the 15 keV and 20 keV processes, the hBN Raman mode suffers a decrease in intensity and a broadening of the FWHM. In particular, the FWHM increases up to 13.6 cm^−1^ and 10.2 cm^−1^ for the 15 keV and 20 keV processes, respectively, indicating a worsening of the crystallinity of hBN [55]. It is worth mentioning that no previous works, to the best of our knowledge, has correlated the appearance of light emitting color centers by low-energy electron irradiation with the hBN crystallinity in terms of the FWHM of the hBN E_2g_ peak [44,45,46]. However, high-energy electron bombardment has been demonstrated to generate a moderate concentration of optically active defects without inducing any broadening of the Raman mode FWHM [48,49]. Two recent works have reported defect-activated Raman modes (analogous to graphene D peak) in hBN flake in the case of ion irradiation [56,57]. In the case of He ion irradiation, Liang et al. have reported the appearance of a broad bump at 1295 cm^−1^, that becomes the dominant peak at an ion fluence of 1 × 10^17^ ions/cm^2^, and Gu et al. have reported a broad band at 1609 cm^−1^ after irradiation of hBN with thallium ions with a fluence of 1 × 10^9^ ions/cm^2^. Both these Raman modes may actually be related to the presence of amorphous carbon, deposited during the irradiation process [58,59]. We discuss in the Appendix A, similar findings in sacrificial areas of the sample, where the electron irradiation causes the deposition of amorphous carbon due to carbon residuals present in the SEM chamber. 

The comparative PL and Raman analyses can give indications about the crystallographic attribution, giving rise to the aforementioned light emissions. The broad band composed of the 610 nm and the 660 nm peaks appears in areas having a high degree of crystalline disorder, as demonstrated by the broadening or the complete disappearance of the Raman peak. Therefore, the possible attributions to crystalline defects can be multiple, however, an intense double peak at around 610 nm with a shoulder at 660 nm has been localized at thickness step in mechanically exfoliated hBN flakes [55]. This emission has been recently attributed to substitutional boron atoms [60] or boron vacancy [61,62,63] in hBN grown with chemical vapor deposition. By comparing these results with theoretical calculations, the possible defects involved in these radiative recombinations may be the substitutional boron or interstitial boron atoms, despite the high formation energy [64]. 

The crystallographic origin of the 535 nm emission has been previously assigned to an array of dislocations in hBN [65]. Analogous electron-irradiation-induced generation of line defects has also been previously demonstrated in the case of MoS_2_ [66]. However, such a radiative recombination has been detected in deformed few-atomic-layer hBN induced by substrate patterning [67] or close to morphological defects in mechanically exfoliated flakes [55].

Generally, the generation of the optical active centers can be ascribed to multiple mechanisms [68]. When irradiating with energies below the knock-off energy of hBN, we expect various mechanisms, that we can group in two main categories: the mechanisms that involve the substrate and the electron–hBN interaction. A possible substrate-involving mechanism is the Coulomb explosion arising from the charge accumulation in the SiO_2_ layer, that is caused by a large number of scattering events occurring in this layer, as we reported in the Monte Carlo simulations [69]. Another possible mechanism related to the substrate is the backscattered electron (BSE) irradiation of hBN. BSEs are generated at the SiO_2_/Si interface and cause an increase in the number of electron–hBN interaction events, as previously reported in case of graphene on SiO_2_/Si substrate [7]. The last possible substrate-related mechanism is related to the generation of highly reactive oxygen species (ROS) due to the electron bombardment of the SiO_2_. The electron-irradiation-generated ROS can induce point defects and even local oxidation of the hBN [70,71]. In the case of the direct effect of the electron irradiation of the hBN flake, the local Coulomb explosion has been previously demonstrated under 300 keV electron irradiation on hBN nanotubes [72]. In addition, a recent work [73] about the low-energy electron irradiation of MoS_2_ monolayer has demonstrated that the formation of vacancies through ballistic energy transfer is possible at electron energies which are much lower than the knock-on threshold, when excitations are present in the electronic system. Instead, we rule out the local chemical etching of the hBN, considering that the irradiated area does not present any change in optical contrast, as shown in Appendix A. 

### 3.3. Effect of the Irradiation Spot Spacing

In order to evaluate the effect of the irradiation spot spacing, and thus the interaction between the single irradiation spots, different areas were irradiated increasing the spacing between the single irradiation spots (decreasing the area dose) for the 10 keV irradiation (Figure 3). The intensity PL map obtained at 660 nm in the case of an irradiation pattern with 200 nm spacing is reported in Figure 3a. The dashed yellow square highlights the primary irradiated 2 μm × 2 μm area. The map reveals that the primary impinging area presents a low PL intensity at 660 nm, while it increases in the region surrounding the irradiated area. It is worth noting that the annular surrounding area is composed of a first ring with high intensity extending ~1 μm around the irradiated area and a secondary ring with faint PL intensity, with a lateral size of 500 nm. A similar scenario occurs in the case of the pattern with 500 nm spacing. In fact, the high-intensity annular area is located around the edge of the primary impinging area, while the central area denoted by low PL intensity is smaller than the primary irradiated area. The secondary ring with low emission has a similar lateral extension. The spatial distribution of the irradiation-induced light emission is different in the case of the 1000 nm spaced pattern. The irradiated area presents the highest emission intensity, with decreasing PL intensity receding from it. However, the emission intensity remains high for the first 1 μm outside of the irradiated area and decreases drastically in a secondary ring with a 500 nm lateral extension. PL spectra of irradiated areas with different spacings were acquired (light red lines, Figure 3d) in order to carry out a more quantitative evaluation of the line shape modifications. The PL spectra acquired in the 200 nm and 500 nm spaced irradiated areas are broad bands peaking at 610 nm, showing an additional component at 660 nm and a broad tail in the short wavelength range. Conversely, the PL spectrum of the 1000 nm spaced area presents a higher intensity in the whole wavelength range with the 610 nm and 660 nm components with comparable intensities. For the sake of clarity, we report the PL spectra (dark red lines in Figure 3d) of the high-intensity annular areas for both the 200 nm and 500 nm spacings. In the case of the 200 nm spacing, the comparison of the irradiated area and its surrounding ring shows that the 610 nm and 660 nm emissions have an intensity increase of 400% and of 580%, respectively. On the contrary, the irradiated region with 500 nm spacing exhibits a lower increase in the emission intensity (330% and 420%, respectively). These data can be interpreted in view of the increasing area dose resulting from the different spacing: the shorter the spacing, the stronger the adjacent spot interactions. In fact, the electron irradiation with an area dose higher than 14 nC/μm^2^ causes a cascade effect that induces the generation of non-radiative recombination paths that quench the light emission in the primary irradiated area. Similar results have been demonstrated, using molecular dynamics simulation, for high fluence ion irradiation, where the high fluence process creates a noticeable amount of complex vacancies, leading to amorphization and ultimately complete destruction of hBN [74]. The presence of the high-intensity rings is due to a lower effective area dose compared to the primary irradiated area, because the material is mainly affected by the irradiation sites on the edges of the irradiated area. The lateral extension of such rings is reliable with the radius of the generation/recombination volume obtained by Monte Carlo simulations (Appendix A).

Similar pattern spacing is used when irradiating with 15 keV, whose PL mapping is reported in Figure 4. The 535 nm PL maps of the 200 nm, 500 nm and 1000 nm spacings are presented in Figure 4a–c, respectively. The PL map of the 200 nm spaced irradiation reveals a similar intensity spatial distribution of the 500 nm spaced 10 keV irradiation, where the primary impinging area has a faint PL intensity at 535 nm, while the surrounding annular area has an intense PL emission with 1 μm extension. It is worth noting that in the case of the 15 keV irradiation the secondary annular area has a larger spatial extension (about 2 μm) with respect to the 10 keV irradiation. This effect also occurs in the 500 nm and 1000 nm irradiation patterns, where the surrounding annular areas have a lateral extension of 2 μm. However, both the 500 nm and 1000 nm spacings show an intense PL emission localized in the primary irradiated area. Comparing the PL intensities in the irradiated areas, the 500 nm spacing is more efficient than the 1000 nm patterning in the generation of radiative centers emitting at 535 nm, reaching the maximum intensity value of 7 kcps. Figure 4d reports the PL spectra obtained in the irradiated areas with different spacings. All the PL spectra report that the sharp emission at 535 nm, with the related phonon replicas, appears in the case of all the different spacings. In the case of the 200 nm spacing, the main difference between the irradiated and the surrounding annular areas is the intensity of the emission, in fact, the line shape shows no peculiar differences. In the case of the 1000 nm PL spectrum, the 660 nm component appears more pronounced than in the other PL spectra. The FWHM maps of the hBN Raman mode for the different pattern spacings are reported in Appendix A and representative Raman spectra are shown in Appendix A in order to demonstrate the absence of amorphous carbon induced by electron beam deposition. 

### 3.4. Excitation Power Dependence of the 535 nm Emission

In order to gain more insights on the nature of the sharp emission at 535 nm, a power-dependent PL analysis was carried out (Figure 5). The PL spectra acquired varying the excitation power of the 473 nm laser are shown in Figure 5a. As expected, the phonon replicas at 550 nm and 575 nm appear more pronounced mainly at higher excitation power. Figure 5b summarizes the power-dependent intensity (black dots) and peak position (cyan dots) of the 535 nm band. The 535 peak intensity presents an exponential monotonic increase with the excitation power. It is worth noting that, even when increasing the excitation power over two orders of magnitude, we were not able to reach the saturation of the emission intensity. Moreover, by considering a double logarithmic scale graph (more details in the Appendix A), indications on the nature of the radiative recombination process can be obtained from the slope of the intensity increase [75,76]. In this particular case, the slope shows a close linear behavior, suggesting that the emission is due to an excitonic recombination bound to an isoelectronic trap [75]. 

Finally, the power dependence of the peak position reveals no shift in the emission. This result gives interesting insights about the nature of the radiative recombination: the emission could arise from color centers generated in the irradiation process. Indeed, the absence of the emission shift rules out the possible attribution to donor–acceptor pair (DAP) radiative recombination [77]. In a DAP transition, the peak position is blue shifted due to the increasing of the Coulomb interaction with increased DAPs at high excitation intensity. Consequently, the power-dependent PL results suggest that one possible electronic transition is from a localized defect level to the valence band or conduction band [78].

## 4. Conclusions

In conclusion, we demonstrate that low-energy electron beam irradiation parameters are crucial for tailoring and tuning the light emission properties of hexagonal boron nitride flakes. The irradiation parameters can give rise to emissions at different wavelengths with different widths. Finding the best trade-off between flake thickness and electron beam energy (15 keV) allows engineering of a sharp emission at 535 nm, attributed to an excitonic recombination bound to an isoelectronic trap. Electron irradiation at a lower energy (10 keV) induces a high degree of crystalline disorder that causes the generation of broad emissions. The position-controlled irradiation permits the creation of irradiation patterns with different spacings with increasing area doses. The area dose increase demonstrates the presence of a threshold value, different for each electron beam energy, which, once exceeded, causes the generation of non-radiative recombination pathways in the primary irradiated area. 

## Figures and Tables

**Figure 1 nanomaterials-13-00739-f001:**
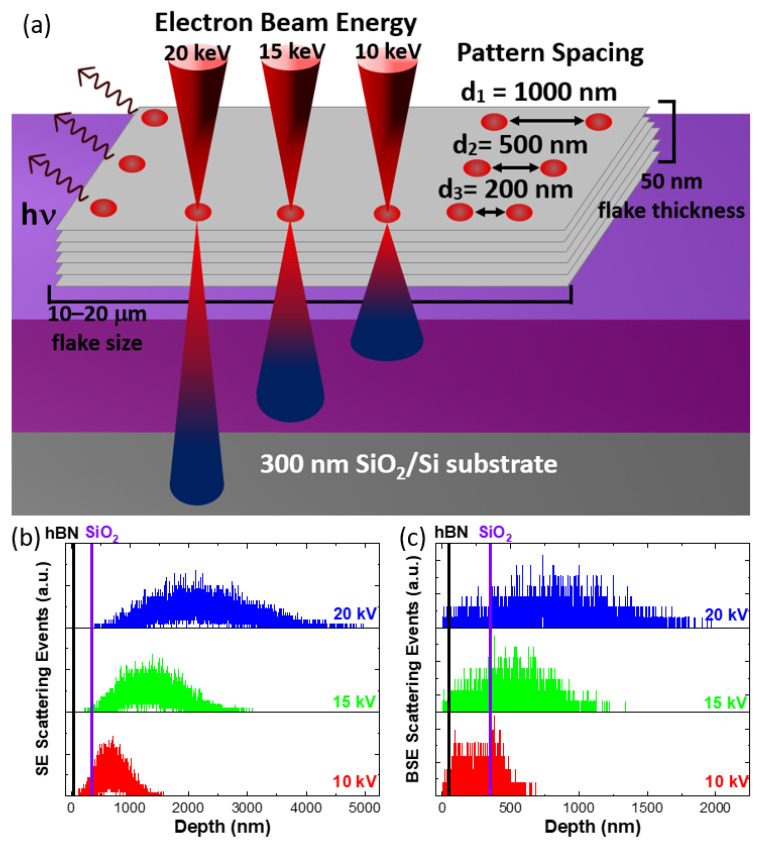
(**a**) Sketch of the different irradiation processes performed on the hBN flakes. Monte Carlo simulation of the depth penetration at different beam energies: (**b**) secondary electron-generating scattering events and (**c**) backscattering electron-generating scattering events. The black and purple vertical lines indicate the thickness of the hBN and of the SiO_2_, respectively.

**Figure 2 nanomaterials-13-00739-f002:**
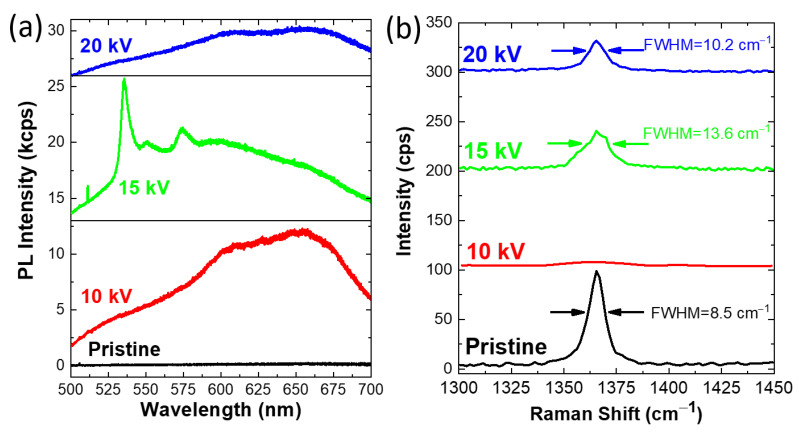
(**a**) PL spectra of the hBN irradiated with different electron beam energies: pristine hBN (black line), 10 keV (red line), 15 keV (green line), 20 keV (blue line) irradiation. (**b**) Raman spectra obtained in the irradiated areas. All spectra are vertically shifted for a better comparison.

**Figure 3 nanomaterials-13-00739-f003:**
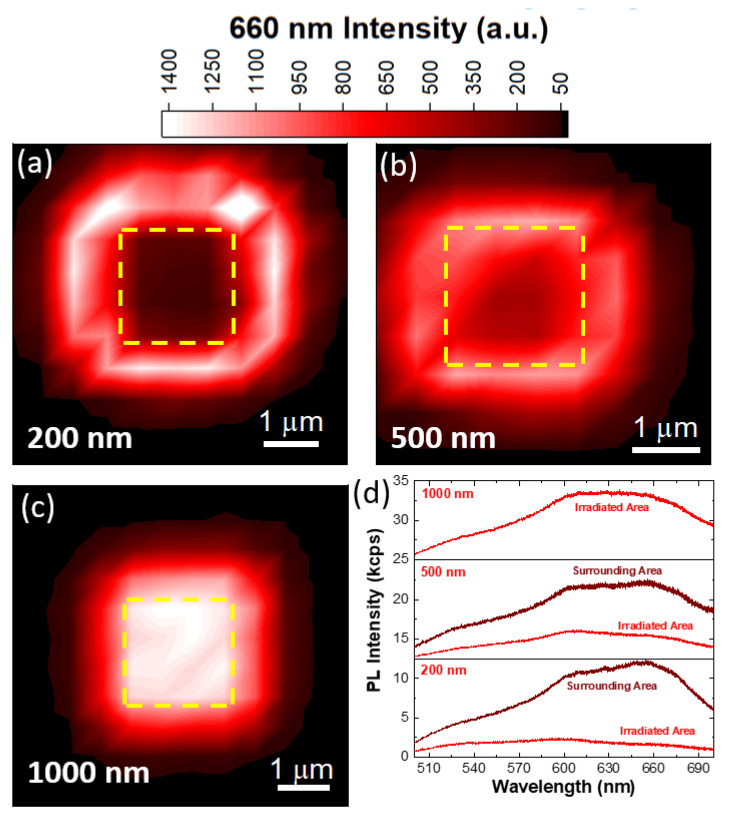
PL intensity map obtained at 660 nm for patterns with 200 nm (**a**), 500 nm (**b**) and 1000 nm (**c**) spacings. The upper color bar applies for all the maps. The dashed yellow square indicates the primary irradiated areas. (**d**) PL spectra of the 10 keV electron-irradiated hBN with different pattern spacings.

**Figure 4 nanomaterials-13-00739-f004:**
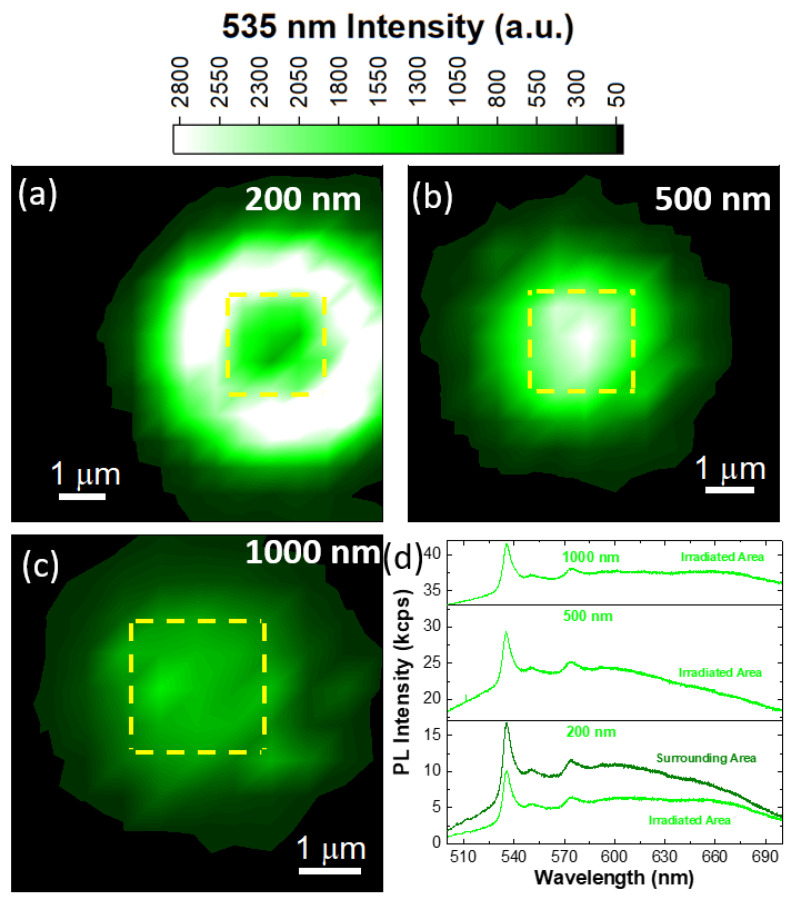
PL intensity map acquired at 535 nm for the 200 nm (**a**), 500 nm (**b**) and 1000 nm (**c**) spaced patterns. The upper color bar applies for all the maps. (**d**) PL spectra of the 15 keV electron-irradiated hBN with different pattern spacings.

**Figure 5 nanomaterials-13-00739-f005:**
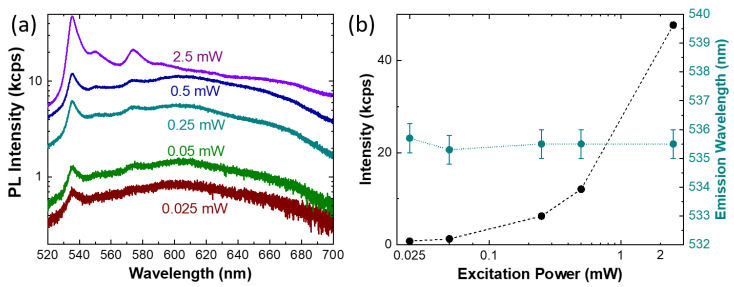
(**a**) PL spectra of the 15 keV electron-irradiated hBN with increasing 473 nm excitation power. (**b**) Power dependence of the 535 nm emission intensity (black symbols) and evaluation of the emission energy (cyan symbols), the lines are only a guide to the eye for the reader.

## Data Availability

The data presented in this study are available on request from the corresponding author.

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
