# Peer review of "Engineering Multicolor Radiative Centers in hBN Flakes by Varying the Electron Beam Irradiation Parameters"

_nanomaterials, 2023, doi:10.3390/nano13040739_

Round 1

Reviewer 1 Report

The Authors report the results of an experimental study where electron irradiation with beam energies in the range of 10-15 keV was used to produce optically active defects in h-BN flakes. The irradiated flakes were characterized by several techniques, and the dependence of the results on beam power, electron dose, flake thickness and other parameters has been investigated. As enormous amount of attention has recently been paid to defect-based quantum emitters in h-BN and other 2D systems, the topic of this study is definitely timely and important. The results are most likely technically correct, and they should present considerable interest to the community. Thus I recommend the manuscript for publication. I would suggest, though, that the Authors address (at least discuss at the level of a working hypothesis) the damage production mechanisms and types of defects which give rise to photoluminescence, which would make the impact of the paper much stronger.

In particular, the Authors should clearly indicate that the electron energies used were well below the knock-on threshold in h-BN, and discuss the possible mechanisms of defect production (electronic excitations, charge accumulation in h-BN or substrate followed by Coulombic explosion, beam-induced chemical etching, etc.). It is also highly desirable to discuss the nature of the most likely defects (substitutional impurities, e.g., carbon, oxygen silicon, vacancies, etc.).

Some other issues:

Lines 167-168. The Authors say: “…the 10 keV irradiation causes a PL intensity that is double the one obtained irradiating 168 hBN at the 20 keV.” This is not obvious from Fig.2a. In fact, the opposite can de deduced from the number on the Y-axis of the plot. I assume for each beam energy the scale starts from zero, and it is not sequential, as shown in the figure. If yes, this should be fixed.

Line 91. The Authors say: “Defects were induced on part of the graphene flakes…” Did the Author actually mean h-BN, not graphene?

Author Response

Dear editor,

We thank the referees for the positive feedback and interesting comments, which have improved the manuscript. After performing new experiments, we have addressed all of them in the attached file. Please see the attachment.  

We look forward to hearing from you,

With kind regards,

Cordially,

Filippo Fabbri

Reviewer 2 Report

This manuscript investigates the multicolor emission characteristics of hBN flakes excited with different electron beam irradiation parameters. The electron penetration depth and scattering events caused by the impinging of electron beam were evaluated by the Monte Carlo simulations. The effects of electron beam energy, irradiation spot spacing, and excitation power on the PL properties of hBN flakes were discussed. The manuscript is acceptable for publication in the Nanomaterials.

The suggestions for manuscript revision are as follows:

1.     The authors should check the values written in the text:

Line 219: …double peak at around 610 nm with a shoulder at 660 nm has…

Line 241: …The intensity PL map obtained at 640 nm in the case of…

Line 244: …low PL intensity at 640 nm, while it…

Line 258: …irradiated areas are broad bands peaked at 600 nm, showing…

Line 261:…whole wavelength range with the 600 nm and 660 nm components…

2.    When an abbreviation first appears in the manuscript, its full name must be presented.

3.    For the “3.2 Effect of the irradiation spot spacing”, the subtitle number is incorrect; please check.

Author Response

(The authors gave the same response as above.)

Reviewer 3 Report

The manuscript present results of the investigation of the electron beam impact in SEM on the photoluminescence of the structure consisting of a few nanometer thin hBN layer on SiO2/Si substrate. The authors claimed that the observed luminescence was originated from the defects in hBN caused by 10-20 keV electron bombardment despite of the fact that the most damages took place in the substrate. However the suggested interpretation of the obtained results gives rise to serious doubts.

1.The presented luminescence spectra are very similar to the cathodoluminescence spectra of Sio2 reported previously ( see H.-J Fitting et al “Cathodoluminescence of crystalline and amorphous SiO2 and GeO2” ,Journal of Non-Crystalline Solids, 279 (2001). 51-59,https://doi.org/10.1016/S0022-3093(00)00348-3;  Baraban, A. P et al «Luminescence of SiO2 Layers on Silicon at Various Types of Excitation». Journal of Luminescence 205 (2019): 102–8. https://doi.org/10.1016/j.jlumin.2018.09.009).

This makes more probable that the luminescence observed in the reviewed paper stems from thee substrate and the measurements on bare substrate regions subjected by the electron bombardment are needed to establish the source of the luminescence.

2. The author did not discuss the impact of carbon contaminations on the photonic properties of the investigated samples , in particular, on the vanishing of Raman signal after the electron beam irradiation. On the other side the formation of carbon-rich film on the sample surface is a very well known phenomenon ( see, for example Dorp, W. F. van, и C. W. Hagen. «A Critical Literature Review of Focused Electron Beam Induced Deposition». Journal of Applied Physics 104,(2008 ): 081301. https://doi.org/10.1063/1.2977587. Such film could have a great impact on the signal both from the substrate and from the thin hBN layer. One should note that the inhomogeneity character of the spatial distribution of the luminescence is somewhat similar to carbon film thickness distribution on the electron irradiates surface ( see fig. 23-24 in the abovementioned paper.)

It is necessary to perform additional measurements of luminescent and Raman signal after the removal of the carbon-rich layer with oxygen plasma.

Author Response

(The authors gave the same response as above.)

Reviewer 4 Report

The authors work is aimed at studying the color centers that are formed by an electron beam in hexagonal boron nitride. At present, the color centers have already been successfully created in this material by electron beams exposed and other types of electromagnetic waves, for example, lasers. Thus, the idea is not new, but the authors presented some rather curious new results. Unfortunately, in this form, the article cannot be recommended for publication in the journal Nanomaterials for the following reasons:

1. The authors used only three accelerating electron voltages, led to nonmonotonic dependences of the resulting luminescent centers properties. In particular, the photoluminescence spectrum of samples irradiated at 15 keV differs from the spectra of samples obtained at 10 and 20 keV. The impact of 10 keV leads to complete amorphization of the sample, and an increase of the electron energy makes it possible to preserve the crystal structure. Thus, the presented results look random, they cannot be systematized and validated.

2. The authors claim that during exposure to an electron beam, carbon is deposited on the sample surface. Samples were prepared with different numbers of dots per 2x2 µm square, i.e. 9, 16 and 100 dots for 1000 nm, 500 nm and 200 nm samples, respectively. Given that each point was irradiated for the same time, it turns out that for different samples, the time of carbon deposition is different. The authors do not provide data on the obtained carbon films thickness, but it is obvious that for a 200 nm sample, the carbon film should be the thickest and can shield the optical signal, which leads to the images in Figures 3a and 4a.

3. How the Raman band width was measured? The spectra on Figure 2a are very noisy, so the reported FWHM data cannot be considered reliable. Moreover, the device spectral resolution is not indicated.

Author Response

(The authors gave the same response as above.)

Reviewer 5 Report

In this work, Authors reported about multicolor radiative emissions, in hBN flakes, engineered by position controlled low-energy electron irradiation. Authors claim that variating the irradiation parameters, such as the electron beam energy and/or area dose, it is possible to induce light emissions at different wavelength in the visible range. In particular, in the orange-red range, and in the green range. Manuscript is appropriate for this journal and I would recommend this work for publication. But I have a few comments:

1) Lines 14-15-“…we are able to induce light emissions at different wavelengths in the visible range.” But in reality Authors demonstrated only two colors (green and red). The sentence should be written more correct

2) Fig. S4 –absent captions for  b and c 

3) Line 91 “Defects were induced on part of the graphene flakes by irradiating with electrons”. Authors probably means hBN flakes

4) Why was chousen the region of electron beam energy from 10 keV to 20 keV? Why did not use also 5 keV and 30 keV?

5) Authors should show the AFM images including irradiated areas and around them as min for 10 keV and 20 keV. It can clear confirm the absence of some carbon contaminants and defects caused by irradiation process

6)  The presented Raman spectra are average or taken from one point? How many points were used for Raman measurements? It would be nice if Authors  can show Raman map of irradiated areas

7) Authors shown the effect of irradiation parameters for a fixed flake thickness. But it is not clear from the work, whether there is a dependence the radiation doses on the thickness of the flakes? What can be expected with flake thicknesses for instance  one–ten layers or 200-400 nm?

Author Response

(The authors gave the same response as above.)

Round 2

Reviewer 3 Report

The revised version of the manuscript is now significantly improved by the authors and can be accepted for publication after some minor changes.

1.     The information about the absence of the impact of electron irradiation on the PL spectrum of bare Si02 is now presented only in Cover Letter in a pared-down form caused by overlapping the figure image and the text. The  full and correct information of this kind has to be included in the main text.

2.     Technical remarks:

Lines 94 and 97 are in fact a repetition of similar data on spot spacing;

Lines 290 and 296 used the adverb “conversely” in the same paragraph that hinders a proper understanding of the discussion;

Scale bar length in Figure S4a fails.

Author Response

Dear editor,
We thank reviewer 3 for the positive feedback and the additional comments. Please see the attachment, reporting the modifications to the manuscript. 

We look forward to hearing from you,
With kind regards,
Cordially,

Filippo Fabbri

Reviewer 4 Report

The authors have made all the necessary corrections to the text and provided comprehensive explanations. The work is recommended for publication in the journal "Nanomaterials" in its current form.

Author Response

We thank the reviewer for his/her work on our manuscript.